# The Value of Natural Stones to Gain in the Cultural and Geological Diversity of Our Global Heritage

Dolores Pereira 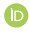

Department of Geology, University of Salamanca, 37008 Salamanca, Spain; mdp@usal.es

**Abstract:** The use of stone to build shelters was an important step in terms of ensuring buildings' durability and security in pre-historic times. It was also an acknowledgment of power and grandeur when societies demonstrated their respect for their leaders and gods by building stone monuments. For this reason, cathedrals, churches, and other magnificent religious monuments were built around the globe; however, the importance of the stone used in their construction itself is not sufficiently appreciated as the key factor ensuring the endurance of these historic buildings. While Western countries have long used iconic buildings to promote their heritage and advance in other socio-economic aspects, including tourism, other regions are yet to take full advantage of this outlook, even though their monumental structures may be equally impressive. Those important monuments are built of stone, which were referred to as Heritage Stones by some research groups, and their recognition would showcase the cultural and geological diversity of the world. However, there are many stones of heritage importance from geographic areas that are under-represented in the scientific literature and the work of research groups. This paper presents a review of the degree of geographical diversity in the recognition of stones and their heritage value.

**Keywords:** natural stone; heritage stone; geographical diversity

## 1. Introduction

Understanding the cultural significance of geodiversity to peoples of the past is a way of developing historical perspectives on the significance of geodiversity [1]. One strategy to highlight the significance of this geodiversity is to designate as many geographic locations as possible as cultural geosites. These locations can include stones that were used for centuries in the construction of stone-built heritage buildings. It must be considered that although natural stone is a noble raw material, it can deteriorate over time due to natural and anthropic events. In recent years, a dramatic increase in the deterioration of the structure of our stone-built heritage was documented as part of the interaction between stone and climatic conditions [2]. Some authors previously predicted the evolution of specific types of lithologies caused by climate change over the next century using different kinds of laboratory experiments and models, concluding that many historic places will experience the deterioration of their historic buildings [3] and their subsequent disappearance if no action is taken to prevent this outcome [4].

Strategies to protect the stone-built heritage are being investigated; however, if replacement of stone is necessary, a major effort should be made to identify the stone that was used originally, ensuring that the replacement does not affect either the aesthetics or the cultural value. This approach is a reality in places where the natural conditions are destroying valuable heritage and the identification of the stone remains obscure [5].

In recent years, a few groups and individuals showed interest in highlighting the stones that represent an important part of humanity's stone-built heritage. They pointed out the importance of weathering, which can lead to the destruction of heritage. In fact, in cases where weathering, or the consequences of natural and/or anthropic events, affect the integrity of the historical buildings and artifacts, restoration or even replacement should be

considered as the only possible way to maintain the heritage. If that happened, choosing the right stone would be crucial. An example of a relevant research initiative is that carried out by the working group on "Natural Stones and Weathering", which is based at the Geological Institute of the Aachen University, Germany. Through applying different techniques and mapping stone weathering, this group investigated monuments, classifying the different weathering forms affecting the stone via a weathering simulation, allowing a prognosis of the weathering behavior of stones in a monument and providing prevention measures. Full information, including about ongoing projects, can be found at https://www.stone.rwth-aachen.de/index.php (accessed on 10 March 2023). Another initiative was carried out by the Heritage Stones Subcommission (HSS), which was established by the International Union of Geological Sciences (IUGS). Initially, this group functioned as a Task Group (Heritage Stones Task Group (HSTG)) that followed the protocols of IUGS (www.iugs.org (accessed on 10 March 2023)), which is the world's largest association encouraging international cooperation and participation in earth sciences.

The first group, based in Germany, is related to the description of weathering processes. The group studied stones and stone-built heritage from regions that are unrepresented in the literature (e.g., Egypt, Turkey, Jordan, and South Korea; see below). Unfortunately, having visited the website, it seems that this group is not very active at present. Thankfully, the literature on stones from such areas and from other individuals and groups increased in the 21st century. The second research group, i.e., the IUGS subcommission, focuses on the study of the recognition of individual stones that have importance in cultural heritage, as well as geoheritage in general. This paper will mainly use the work performed by the second group to address the proposed objective: achieving global cultural geodiversity.

The Global Heritage Stone Resource (GHSR) concept originated in 2008 as an initiative within Commission 10 on Building Stones and Ornamental Rocks of the International Association for Engineering Geology and the Environment (IAEG), which was formed at the 33rd International Geological Congress in Oslo, Norway. It was the precursor of the working group on heritage stones, which, in 2012, became a formally designated working group (Heritage Stone Task Group—HSTG) within the IUGS [6]. The specific goal of HSTG was to facilitate formal designation of those natural stone types that achieved widespread recognition in human culture, as well as to create the "Global Heritage Stone Resource" (GHSR) as a term for this designation. Stones that were used in heritage construction, sculptural masterpieces, and utilitarian (yet culturally important) applications are obvious candidates. In association with this aim, there was a need to promote the adoption and use of the GHSR designation by international and national authorities. HSTG committed to maintaining a register of GHSR approved stones [6]. The initiative started as a step towards improving knowledge of natural stones that had historical uses and gained international importance. In order to nominate a stone for designation, there is a list of features that should be documented: names are very important, i.e., both the proposed name for designation and the stratigraphic or geological name, as well as any other name given to different types or variants of the studied stone. Moreover, commercial names used to market the stone are important details used to find the stone in papers, web sites, and/or catalogues. Other important information includes the geographic area where the studied stone occurs in nature, including a map for better location; locations of quarry or quarries (sites of active and abandoned quarries of the stone); the geological age and geological setting, with details of sedimentary basin/fold belt, tectonic domain, igneous activity etc. that place the stone in a wider geological perspective; the petrographic name, with detailed mineralogical description; the primary colour(s) and aesthetics of the stone; the natural variability; the geotechnical properties, including (at least) water absorption, density, porosity, compressive strength, flexural strength, and any other properties that characterize the stone; the suitability and assessment on utilization, such as cut building blocks, sculpting stone, roofing, monuments, polished decorative use, technological objects, etc.; the vulnerability and maintenance of supply, which is used to know if the stone is available for future supply; the historic use and geographic area of utilization; heritage

utilization, including an extensive list of significant buildings, monuments, sculptures etc, including dates of construction; and related heritage issues and information on other stones that can be geographically related. All this information should be accompanied by principal literature dealing with the stone to reflect on the importance of the stone being nominated for IUGS designation [6].

The HSTG was promoted to become a subcommission within the IUGS in 2016: the HSS. Together with the Geosites Subcommission, it constitutes the International Commission on Geoheritage (ICG) (Figure 1).

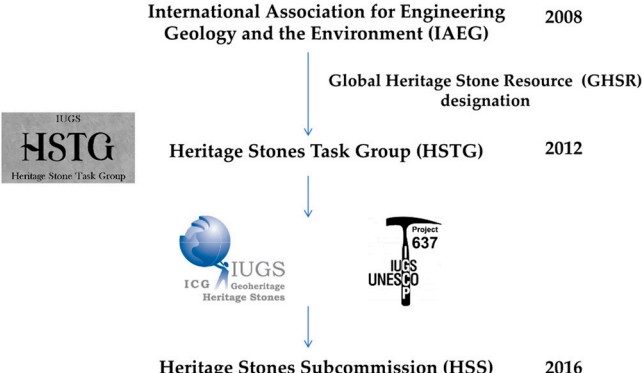

**Figure 1.** Evolution of Heritage Stones working group, since its creation as a IUGS Task Group, in 2012, until today, as IUGS Heritage Stones Subcommission (HSS).

The main target of HSS was to discover stones with great cultural value due to being part of important monuments and historic buildings. Sometimes, these stones were only recognized locally or regionally through publications in obscure journals that either had no impact within the scientific community or were published in a language that was not English (e.g., Swedish, Russian, Chinese, Turkish, etc.). A large community of scientists, who are experts on natural stones and their cultural value, joined the HSS and started a list of potential candidate stones that should be recognized as cultural symbols (see Table 1 in the Section 2). For that purpose, the IUGS recognized a geological standard: the Global Heritage Stone Resource. The researchers involved in the subcommission gradually started to publish studies in prestigious international journals that discussed the characteristics of stones that previously remained hidden in local publications (e.g., [7,8] and references therein). Such was the case of the Kolmården serpentine marble, which is a very interesting stone that was studied by geologists from the Swedish Geological Survey [9], who only published the results in Swedish. Another example is the Russian red Shoksha quartzite, which ended up as the tombstone of Napoleon, Russia's great enemy [10]. During the period lasting from 2013 to 2020, an interim list of 90 potential heritage stone candidates was created (see Section 2). All those stones were subject to research projects that published their results and presented their outcomes at international conferences, where the working group discussed the progress of the work and how to improve transparency through including stones from as many countries as possible. The evolution of the process of recognition of stones in heritage was part of the UNESCO IGCP-637 initiative [11], and all the information was stored on a website. Starting in 1972, UNESCO developed the International Geoscience Programme (IGCP), which serves as a knowledge hub to facilitate international scientific cooperation in the geosciences. The IGCP mission promotes sustainable use of natural resources, advances new initiatives related to geo-diversity, geoheritage, and geohazard risk mitigation (https://en.unesco.org/international-geoscience-programme (accessed on 15 May 2023)). Therefore, the recognition of the Heritage Stones working group within the IGCP program impelled the promotion of heritage stones at an international level [11].

**Table 1.** Original list of potential candidate stones for designation as Heritage Stones created by HSS at its founding point. Source: https://diarium.usal.es/mdp/heritage-stone-task-group/ (accessed on 15 April 2023).

| Country | Name of the Stone | Lithology | Continent |
|---|---|---|---|
| Albania | Dropulli Stone | Limestone | Europe |
| Argentina | Piedra Mar del Plata | Orthoquartzite | South America |
| Armenia | Armenian Tuff | Volcanic tuff | Asia |
| Australia | Sydney Sandstone | Sandstone | Oceania |
| Australia | Victorian Bluestone | Basalt | Oceania |
| Australia | Harcourt Granite | Granite | Oceania |
| Australia | Mintaro Slate | Slate | Oceania |
| Australia | Austral Black Granite | Norite | Oceania |
| Australia | Mount Gambier Stone | Limestone | Oceania |
| Australia | West Australian Coastal | Limestone | Oceania |
| Austria | "Rot Scheck" Marble | Marble | Europe |
| Brazil | Marmore Branco Paraná | Marble | South America |
| Brazil | Minas Gerais Schist | Schist | South America |
| Brazil | Ornamental Soapstone | Talc/Serpentine | South America |
| Belgium | Belgian Red Marble | Marble | Europe |
| Belgium | Lede Stone | Sandstone | Europe |
| Belgium | Petit Granit (Bluestone) | Limestone | Europe |
| Belgium | Tournai Marble | Marble | Europe |
| Canada | Tyndall Stone | Dolomite | North America |
| Finland | Rapakivi Granite | Granite | Europe |
| Finland | Tuulikivi | Soapstone | Europe |
| France | Meulière de Brie | Siliceous rock | Europe |
| France | Pierre de Caen | Limestone | Europe |
| Germany | Solenhofen Limestone | Limestone | Europe |
| Greece | Pentelikon Marble | Marble | Europe |
| Greece | Thassos Marble | Marble | Europe |
| Hungary | Hungarian Travertine | Travertine | Europe |
| Italy | Carrara Marble Province | Marble | Europe |
| Italy | Gravina Calcarenite | Sandstone | Europe |
| Italy | Italian Slate | Slate | Europe |
| Italy | Lecce Sandstone | Sandstone | Europe |
| Italy | Pietra Serena | Sandstone | Europe |
| Italy | Roman Travertine | Travertine | Europe |
| Italy | Rosa Beta Granite | Granite | Europe |
| Italy | Rosso di Verona | Marble | Europe |
| Italy | Trani Limestone | Limestone | Europe |
| Italy | Trentino Porphyry | Porphyry | Europe |
| Japan | Hiroshima Granite | Granite | Asia |
| Japan | Koto Rhyolite | Tuff | Asia |

**Table 1.** *Cont.*

| Country | Name of the Stone | Lithology | Continent |
| --- | --- | --- | --- |
| Japan | Komatsu Stone | Andesite | Asia |
| Malta | Globigerina Limestone | Limestone | Europe |
| New Zealand | Oamaru Stone | Limestone | Oceania |
| Norway | Blue Pearl Granite | Larvikite | Europe |
| Norway | Otta Schist | Slate | Europe |
| Norway | Oppdal Schist | Schist | Europe |
| Norway | Norwegian Rose | Marble | Europe |
| Norway | Hove/Porsgrunn | Limestone | Europe |
| Portugal | Estremoz Marble | Marble | Europe |
| Portugal | Lioz limestone | Limestone | Europe |
| Portugal | Arrábida Breccia | Breccia | Europe |
| Portugal | Oporto Granite | Granite | Europe |
| Romania | Ruschita Romanian Marble | Marble | Europe |
| Slovenia | Karst Limestones | Limestone | Europe |
| Slovenia | Podpec Limestone | Limestone | Europe |
| South Africa | Impala Granite | Norite/Gabbro | Africa |
| Spain | Alpedrete Granite | Granite | Europe |
| Spain | Azul Platino | Granite | Europe |
| Spain | Colmenar Limestone | Limestone | Europe |
| Spain | Crema Marfil | Limestone | Europe |
| Spain | Negro Markina | Limestone | Europe |
| Spain | Piedra Dorada | Limestone | Europe |
| Spain | Piedra Pajarilla | Granite | Europe |
| Spain | Rosa Porriño | Granite | Europe |
| Spain | Sierra Nevada Serpentinite | Serpentinite | Europe |
| Spain | Villamayor (Golden) Stone | Sandstone | Europe |
| Spain | Santa Pudia Calcarenite | Limestone | Europe |
| Spain | White Macael Marble | Marble | Europe |
| Sweden | Älvdalen Porphyry | Porphyry | Europe |
| Sweden | Kolmarden Serpentine Marble | Serpentine | Europe |
| Sweden | Hallandia Gneiss | Gneiss | Europe |
| Sweden | Black Granite | Diabase/Dolerite | Europe |
| Sweden | Bohus Granite | Granite | Europe |
| Sweden | Ölandskalksten | Limestone | Europe |
| Sweden | Vånga | Granite | Europe |
| Switzerland | Marmore von Grindelwald | Marble | Europe |
| United Kingdom | Ailsa Craig Granite | Granite | Europe |
| United Kingdom | Bath Stone | Limestone | Europe |
| United Kingdom | Granite from Devon | Granite | Europe |
| United Kingdom | Purbeck Stone | Limestone | Europe |
| United Kingdom | Portland Limestone | Limestone | Europe |

**Table 1.** *Cont.*

| Country | Name of the Stone | Lithology | Continent |
|---|---|---|---|
| Japan | Komatsu Stone | Andesite | Asia |
| United Kingdom | Welsh Slate | Slate | Europe |
| United Kingdom | Derbyshire Black Marble | Marble | Europe |
| United States | Athens Marble | Dolomite | North America |
| United States | Barre Gray Granite | Granite | North America |
| United States | Indiana Limestone | Limestone | North America |
| United States | Pennsylvania Slate | Slate | North America |
| United States | Jacobsville Sandstone | Sandstone | North America |
| United States | Tennessee Marble | Limestone | North America |
| United States | Yule Marble | Marble | North America |

The working protocol of the creators of the HSS was initially focused on soliciting and approving citations for GHSR status. As a first step, the interim or standing list of potential GHSRs continued to grow (see Table 1 in the Section 2). It was, thus, necessary to develop citations together with an essential research paper advocating for the recognition of GHSR in each case. As an example to follow, a research paper on "Portland Stone" from the United Kingdom was published as a model citation for further reference and consideration for possible adoption [12]. Following satisfactory documentation, HSS Terms of Reference advised that recognition of GHSR status had to be formally approved by the HSS Board of Management. The Board was encouraged to consult national or regional authorities and Corresponding Members with respect to draft citations and to revise draft citations as appropriate if deemed necessary. Although the focus was placed on stones used in heritage buildings, newly available dimension stone resources were also considered potential heritage stones if they could meet the necessary criteria in the future. Using this method, many types of dimension stones coming from many different locations might eventually be categorized as a type of heritage stone. A long-term HSS goal at the time was the preparation of an "International Guide to Heritage Stone Designation". This goal would address the intersection area between geological sciences and human culture, as it would focus on both the place of natural stone extraction and the final use of the extracted resource [6].

The work of the HSS was disseminated through a website that was frequently updated by the HSS Secretary General, which provided information, not only regarding the stones that the HSS was studying, but also regarding other relevant publications, conferences, and meetings. Many researchers learned of the existence of this activity through the website and joined the Subcommission, thus increasing the possibility of adding to the list of potential heritage stone candidates. The IGCP-637 played a key role in communicating the activities and achievements of the HSS: by the end of the project, in 2020, 121 researchers from 27 countries (including 15 European countries) had actively collaborated by submitting proposals, reviewing proposals, and publishing papers containing information that would otherwise not be accessible to most of the international community. At the end of the first HSS period, 22 stones were recognized based on the IUGS standard.

In 2020, a new executive took charge of the HSS and a new protocol for nominating important natural stones was implemented [13]. This protocol eliminated the "Global" designation, which made sense, because some stones are only of very high interest on a local scale. The new protocol included the submission of proposals, the review of the proposals, and the nomination of stones by board members only, instead of opening a call for proposals from all correspondents. The result, at the end of the second batch of nominated stones, in 2022, was ten more samples from the countries of the proposers: India, Mexico, Spain, Germany, France, Ireland, and Canada (https://iugs-geoheritage

.org/subcomission-onstones/ (accessed on 10 April 2023). There was also an advance in diversity through including two stones from North America (Teozantla Tuff from Mexico and Tyndall Stone from Canada) and three stones from India (Deccan Basalt, Jaisalmer Limestone, and Alwar Quartzite). Consequently, through introducing these stones, only a slight difference in geographical diversity was included in the world's representative stones. When the new executive took charge of the subcommission, very important information related to the origin of the working group, the initial diversity goals and the working interim list disappeared from the contents of the new website. For this reason, and to retain the historical background of the subcommission, the author of this paper created a space within her own webpage to maintain the information (see below).

The aim of this paper is to review the status of the recognition of stones that play a key role in stone-built heritage and identify a possible way to enrich the geographical diversity of such recognition. This article will analyze potential biases in the process based on the composition of the decision board, which consisted mostly of European researchers, and consider how many stones and countries were ignored. This approach will highlight the unbalanced representation of regions with stones in the interim list, which excluded important regions in Asia, South America, and Africa, where significant yet exotic stones are quarried and exported internationally. Examples of the latter are Angola Black, Moçambique Black Granite, and Namibia Blue Granite. None of them are granites. The black stones are gabbro or other mafic lithologies. The blue one is commercialized as granite, but it is in fact a foidolite of Precambrian age, being made up of big crystals of blue sodalite, similar to the Azul Bahía "granite" quarried in Brazil (Figure 2). All the rocks are exotic and beautiful and well marketed by stone companies (an Italian company, in the case of Namibia Blue, and Chinese companies in the case of gabbro; however, no scientific information is published in journals indexed in the JCR (Journal Citation Reports). This study is an important step towards the recognition of stones that are symbols of their regions. Moreover, it explains the importance of plate tectonics, which created one of the blue stones found on each side of the Atlantic Ocean.

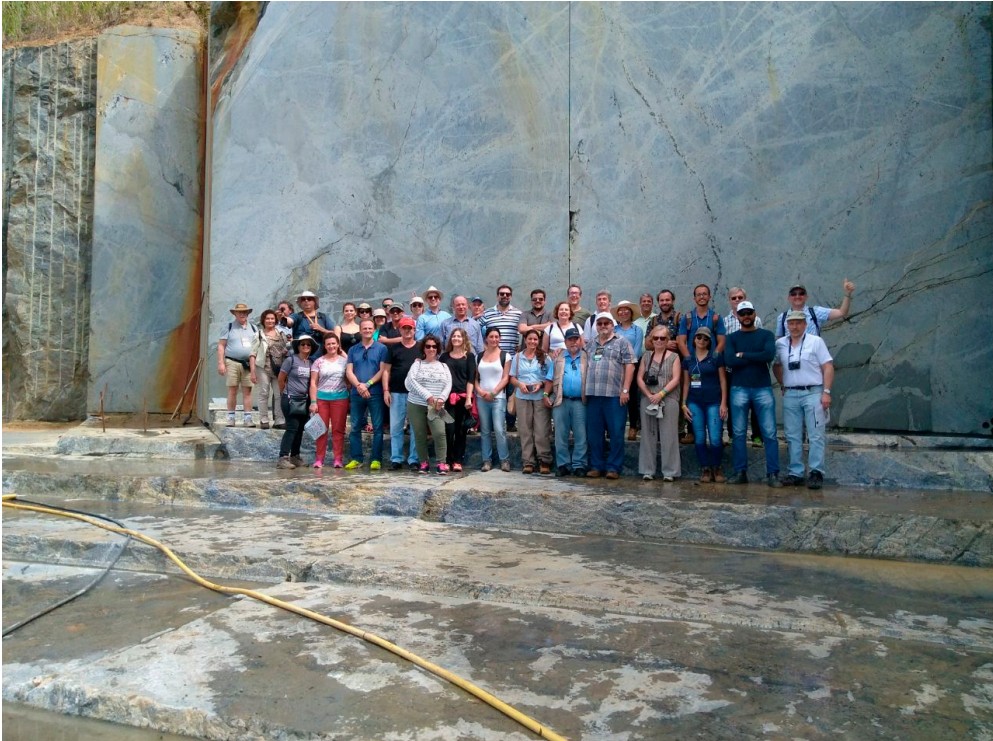

**Figure 2.** Azul Bahía (Bahia Blue) quarry with HSS participants in Global Stone Congress in Ilheus, Bahía, Brazil, in 2018.

This work is not intended to be an exhaustive study of natural, heritage stones from around the world. It is impossible to include all the regions that have merits for such recognition; however, the author hopes that this research will raise the interest of the scientific community working on stones in the most under-represented regions.

## 2. Methodology

To begin this review, the author gathered all the information related to the Heritage Stone working group, extending from its establishment in 2012 to the present day.

The Heritage Stones group started an interim list to widen the typologies, countries, and regions represented (Table 1). This table provided a basic tool to understand the scope of this review, as well as to identify the many geographical gaps that this list contained in order to help researchers to reflect on the many important places and stones that still lack recognition. The author consulted different documents and works on heritage sites where the stone was an important construction material, as well as papers on natural or human-induced destruction of stone-built heritage.

## 3. Discussion

Originally, all the working documents of the HSS were available via the group's web-site. However, when the new board took charge of the management of the group in 2020, the original website disappeared from the Internet, together with the historical information about the group's activities. Most of that information was published in the form of newsletters, scientific papers, and reports, which were collected for this analysis. The website of the IGCP-637, which was the group's engine at the starting point, is no longer available; however, these data and those of the IGCP were moved by the author of this paper, meaning that most of the information can now be accessed at http://diarium.usal.es/mdp (accessed on 15 April 2023).

Based on all available information, this section will discuss the importance of diversity in the recognition of stones in the context of heritage, as well as consider how to involve the under-represented areas through publishing the research on the stones in high-impact publications.

### 3.1. Geographic Diversity

Most data for the present paper came from the first period of the HSS, when more than 120 experts collaborated not only to propose potential candidates, but also to review those proposals in accordance with the HSS terms of reference. The re-established subcommission changed the statutes and the stones are now proposed, reviewed, and voted on by the executive and voting members, including the chair and the secretary general of the subcommission: https://iugs-geoheritage.org/subcomission-on-stones/ (accessed on 15 April 2023).

When creating the working list created, the HSS initially deemed it a good idea to include as much geographical diversity as possible. However, at present, the author of this review can critically state that not only was the list biased (Table 2, Figure 3. Out of 27 locations, 18 were from Europe), but the stones selected for recognition were, and still are, also biased (Figure 4).

Going back to the interim list, even though the HSS members tried to obtain the representation of most regions around the world, given members' expertise and their ongoing projects, some areas were under-represented: Europe had overwhelming regional representation, constituting 71% of the stone candidates in the list; Africa had only one potential stone and Oceania had nine potential stones. None of these 10 candidates were recognized as heritage stones; however, some publications on them were previously released [14,15], which shed light on some areas.

**Table 2.** Geographic areas represented in HSS interim list.

| Region | Number of Stones | % |
|---|---|---|
| Europe | 64 | 71 |
| Oceania | 9 | 10 |
| North America | 8 | 9 |
| South America | 4 | 4.5 |
| Asia | 4 | 4.5 |
| Africa | 1 | 1 |

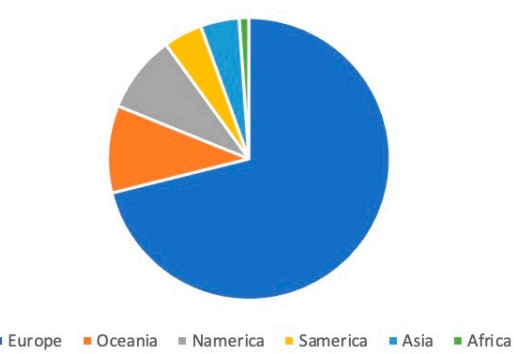

**Figure 3.** Pie chart showing regions with candidate stones in interim list. Remarkable unbalance is clearly shown in chart, with Europe being overrepresented in list.

There is also a lack of diversity in lithology: Limestone was by far the most popular candidate, with almost 29% representation, followed by marble and "other", granite, slate, and sandstone (Table 3, Figure 5). The "Other" category includes porphyry, travertine, serpentinite, gneiss, larvikite, quartzite, norite, siliceous rock, soapstone, basalt/diabase, and tuff.

**Table 3.** Lithologies represented in HSS interim list.

| Lithology | Number of Samples | % |
|---|---|---|
| Limestone | 26 | 28.9 |
| Marble | 17 | 19 |
| Granite | 14 | 15.5 |
| Slate | 9 | 9.8 |
| Sandstone | 7 | 7.8 |
| Other | 17 | 19 |

*3.2. Lithology Diversity in Stones Listed and Recognized by the IUGS*

The IUGS recognition of the stones is supported the fact that limestones are the most studied and used natural stones in heritage (Table 3): 10 out of 22 stones in the first IUGS recognition and 12 out of 32 stones in the second recognition were limestone (https://iugs-geoheritage.org/designations-stones/ (accessed on 15 April 2023)). It must be noted that some of the stones designated as marbles are also limestones, albeit without any metamorphic influence or recrystallization. This finding is another important issue related to the proper characterization and recognition of stones, as marble has a different physical and mechanical response in construction compared to limestone [16–18]; therefore, the importance of recognizing the problems derived from the incorrect naming of stones

is clear. The weight of sedimentary stones in the list makes sense, as sedimentary rocks are the most abundant rocks on Earth's surface, followed by igneous rocks. Granite and basalt are the most common igneous rocks on Earth, although not many basalts were described as building stones, let alone heritage stones. An additional problem is that the ASTM standard specification refers only to granite, which has a very different physical and mechanical characterization than mafic igneous rocks. That problem shows why a wide variety of stones are clustered under the term "other" in the tables above. Ref. [19] explained how ancient resourced, such as the basalts of Sardinia, can still be in use and should be preserved. Ref. [20] described the deterioration of a heritage site in an ancient fort made of basalt. Basalt is a light material with very good mechanical properties that was used to build historical sites in specific regions; however, granites (or granite-like materials) were the only stones that were recognized as potential heritage stones in the working list. Few volcanic rocks are in the present list of IUGS-designated stones (e.g., tuffs from Germany and Mexico (https://iugs-geoheritage.org/designations-stones/ (accessed on 15 April 2023))), which is a step forward in enriching the diversity of lithologies. An effort should be made to point out the differences within the groups, (e.g., separating granites from other igneous rocks, such as gabbro, norite, syenite, etc., as well as explaining the importance of differentiating serpentinite from marble, as serpentinite is listed in some webpages as "green marble" [21], and limestones from marbles, as mechanical properties can be very different (see above)). The factors driving recognition of stones should also be related to their use in the restoration of heritage. For this reason, a proper characterization is needed, because the durability of the stone is directly related to the durability of the stone-built heritage.

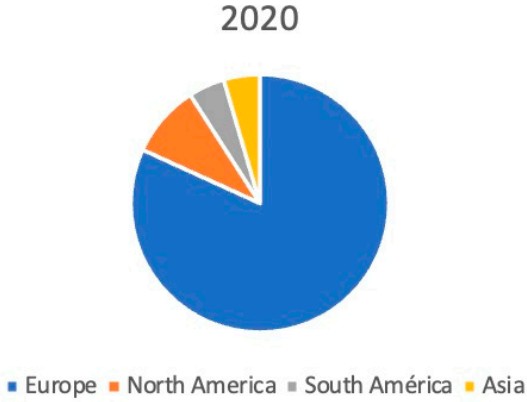

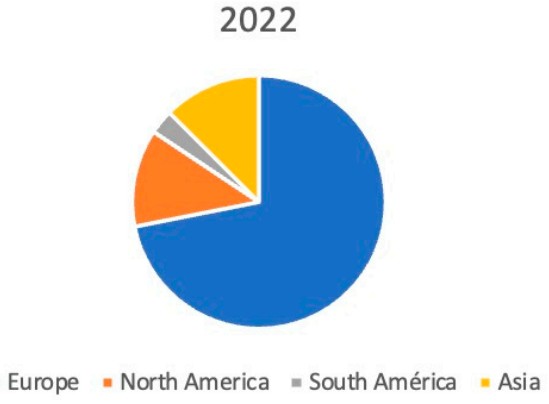

**Figure 4.** Recognized (IUGS designated) heritage stones in 2020 (**above**) and 2022 (**below**). Once again, imbalance is clearly shown in diagram, with Europe being overrepresented in list.

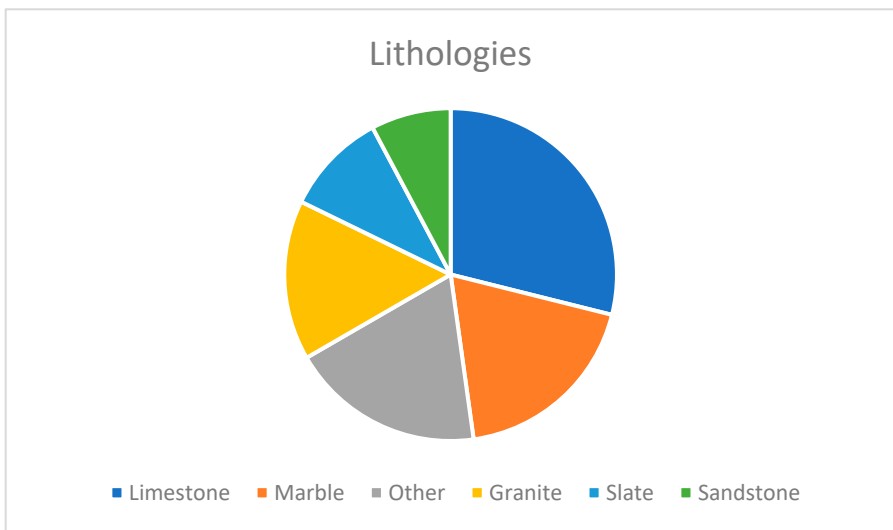

**Figure 5.** Pie chart showing lithologies represented by stones in interim list. "Other" category includes porphyry, travertine, serpentinite, gneiss, larvikite, quartzite, norite, siliceous rock, soapstone, basalt/diabase, and tuff.

## 4. Discussion

Many types of natural stones were used to build heritage buildings all over the world. However, although the architecture and historic buildings of all continents are equally important, the so-called "occidental countries" pushed for the promotion of European heritage and its stones. For this reason, the working lists of research groups working on stones are mainly composed of European examples, as demonstrated above.

Many natural stones and their quarrying are related to cultural and geological sites. A long list of examples can be taken from the Geoparks (e.g., Limestone in Thorsberg Quarry, Platåbergens Geopark, Sweden; the White Limestone Quarry of Mátraszőlős, Novohrad–Nograd Geopark, Hungary–Slovakia; Soapstone at Trollfjell Geopark, Norway; https://en.unesco.org/global-geoparks (accessed on 1 April 2023)). Once again, an imbalance in the recognition of Geoparks is found, as there are only two Geoparks in Africa. Most Geoparks are located in Asia, South America, and Europe [22], and only two of them are found in Africa: M'Goun Global Geopark, in Morocco, and Ngorongoro Lengai, in Tanzania. However, there are prominent heritage sites in Africa. For example, the most impressive sites in Egypt are related to the quarrying and use of natural stone and are part of internationally recognized geoheritage (https://whc.unesco.org/en/list/86/ (accessed on 15 April 2023)). Previous studies showed that the monuments of the Fourth Dynasty of the Giza Plateau are built using limestone from that area [23,24]; those stones should be part of a global list of important stones [25]. More examples of under-represented stones and geographic areas outside Europe are to be found in Asia, where important stones are examples of heritage that needs protection. Marble from Vietnam are well known in the construction sector. They receive several names (e.g., Snow White, Galaxy White, Crystal White, and Wood White). This white marble is made up of a granoblastic mosaic texture, where individual calcite crystals meet other crystals with somehow indented boundaries, which resemble the well-known Carrara marble and have potential implications in the restoration market [26]. Marble from Vietnam is not as scientifically well known as its Italian counterpart. Information can be found on the websites of companies operating in the stone sector; however, to the author's knowledge, no scientific publications link these marbles to local heritage. Stones from Africa, Oceania, and other Asian areas, such as China and the Near and Middle East, remain unacknowledged in recognized scientific networks, even if European, Arabic, and Chinese architectural styles all emerged from long ancient civilizations. The stone-built heritage in countries such as Turkey [27,28], from which both Pavonazzetto marble and Urfa Limestone are published and described

as candidates, have not yet been considered as important heritage stones in heritage; the heritage of Lebanon [29,30], Syria [31], Iraq [32], Israel [33], Jordan, [34], Palestine (where natural stone played an important role in the architecture of old cities in the mountain region, such as Jerusalem, Hebron, Bethlehem, and Nablus) [35,36], Saudi Arabia, other nations of the Arabian Peninsula [37], and Iran [38] is also as important as the heritage of the so-called occidental world. Unfortunately, some of these unrepresented areas are very unstable, with political conflicts (e.g., Syria, Iraq, and Palestine), natural hazards, such as earthquakes (e.g., Turkey), and anthropic hazards, such as wars and accidents (e.g., Taliban wars in Afghanistan, a gas explosion in Lebanon), frequently affecting communities. These problems led to the destruction of important stone-built heritage. Rebuilding the lost heritage can be a challenge, particularly when the stone that was originally used is not specifically identified. Most of the above references are related to the heritage of places, though very few are dedicated to the stones used in their construction.

The main goal of the HSS group, which is a subcommission of the world's largest association in earth sciences (IUGS), was to recognize stones that had been used for centuries to build historic buildings and artifacts that are now World Heritage Sites. However, it was noted that the trajectory of recognition did not change over the years since the creation of the group, and the goal of gaining diversity in geography was not met. Europe is a leader in natural and heritage stones research, yet so many cultures passed through, leaving behind their vernacular architecture, with some buildings built in stone and most recognized as heritage. The most plausible explanation is that Europeans probably had a privileged position in promoting their heritage and the natural stones used to build that heritage and recent constructions. Moreover, there is a clear correlation between the composition of the decision boards of research groups and the lack of diversity of regions and recognized stones, as explained in this paper. This issue is shown in Table 1, the original working document, and the stones finally designated by the IUGS, which mainly come from Europe, with a few coming from the Americas and Asia, and none coming from Oceania and Africa. Moreover, all the stones from Asia that were designated as heritage stones come from India. However, though many stones originating from elsewhere in Asia and Africa deserve recognition, as explained in the text above, they were not mentioned in the interim list created by the HSS at its creation. Important cultures based in Asia, Africa, and South America left a very important heritage, comparable to that of the Europeans; however, so far, they have not been able to promote it within the global earth and material sciences community. The original goal of the Heritage Stones working group, which was to implement diversity, may have not been achieved; however, it triggered the publication of articles in highly cited journals that will raise awareness of these treasures. Heritage and stones were first mentioned together, in the context of the present paper, in 1992 [39]. These authors were concerned about the deterioration that some stones could suffer, thus jeopardizing our heritage. If we now search for the keyword "heritage stones" on the Internet, we can find 349 articles published in a variety of journals (e.g., Heritage, Geological Society Special Publication, Geoheritage, Episodes, Journal of Conservation Science, Journal of Cultural Heritage, Geoscience Canada, Construction and Building Materials, International Biodeterioration and Biodegradation, Applied Sciences, and Coatings). If we search for "Global Heritage Stone", we can find 56 articles published in journals such as Heritage, Geological Society Special Publication, Geoheritage, Episodes, Sustainability, Energy Procedia, Geoscience Canada, Environmental Earth Sciences, and the Arabian Journal of Geosciences (https://exaly.com/, accessed on 28 April 2023). Some of those papers were published in 2022, at which point the term "Global Heritage Stone" had become obsolete, due to the controversy over stones that, although very important for some specific cultures, were only used locally and regionally, even if that use lasted for centuries. The author of this paper noted an increase in publications on stones in recent years. Unfortunately, some of them are not high-quality papers and do not show the notorious work that goes into the characterization of a stone, i.e., study of mineralogy, geochemistry, and physical and mechanical characteristics. However, many of these articles contain only a "copy-paste"

of the characteristics provided by the company's commercial website, which markets the stone in question. Even if this information is useful, the work of a petrologist in the field of natural stones must be original, as it has been shown that some of the data provided on websites are not as accurate as is desirable. As discussed above, limestones are sometimes commercialized as marble, and gabbro, norite, etc., are most times commercialized as "black granite". The proper characterization is very important to understand the deterioration process that some stones can suffer in order to neutralize the negative consequences when ancient historical buildings are affected [5] or when restoration, or even replacement, is necessary, due to natural and/or anthropic causes. Anthropogenic actions can also affect stone-built heritage; thus, if it is to be restored, or even replaced, adequate knowledge of these stones is essential. One recent example is the damage caused by the double explosion that occurred in the port of Beirut on the 4 August 2020. The magnitude of the explosion, which was comparable to a 3.5 magnitude earthquake, devastated the port infrastructure and the surrounding neighborhoods, resulting in loss of life and heritage. The important tangible and intangible heritage of the city was also affected. Some of the most historic neighborhoods suffered severe damage and UNESCO soon showed its concern for the rehabilitation of historic buildings and urban areas, which are essential to revive the cultural heritage of this historic city. Original materials and appropriate techniques must be used for the reconstruction and restoration of buildings in order to preserve Beirut's urban heritage [40]. Lebanese vernacular architecture used stone for a significant part of the city's buildings. Historic buildings in Beirut follow traditional Mediterranean construction methods, using a yellow sandstone for the major construction work and a dolomitic limestone for the ornaments, which were located mainly on the outer façades and obliterated due to the blasts. The immediate action to preserve all the stone pieces that were uprooted due to the explosion and the law that prohibited any kind of full or partial demolition in the affected area, which consisted of a large number of stone buildings (Law No. 194, dated 16 October 2020 [41]), are important factors in the restoration of the historic buildings and the restoration of Beirut to its original urban landscape. The quarries of those local stone materials were identified, the geology of the formations were recognized, and the technical details of the rehabilitation works were gradually published [42–44]. Carrara marble was identified in the structures to be restored, though the most common materials, i.e., limestone and sandstone, are referred to as "local sandstone" and "local limestone". Characterization and recognition of those stones will trigger the complete and appropriate restoration of the affected area and the preservation of heritage [45]. As explained above, the inappropriate, non-scientific characterization and naming of stones can lead to incorrect restoration actions.

## 5. Conclusions

One of the main conclusions is that interest in natural stones and their importance in heritage increased over the last ten years, probably due to the creation of specific and thematic research groups (e.g., HSS). Another conclusion is that it is necessary to include stones from unrepresented areas that also have a rich heritage. Otherwise, the original IUGS initiative can be interpreted as a tool to select elite stones and give them more recognition. The White Marble from Vietnam, the Blue Foidolite from Namibia, and the Urfa Limestone from Turkey are examples of those unrepresented areas, though hundreds of them deserve a place in the literature and recognition of their historical use, whether on a global, regional, or local scale. The additional achievement will be the use of appropriate stones in the restoration process when a monument or a historical building is in danger of deterioration or disappearance, whether due to natural or anthropogenic hazards. In order to preserve heritage, it is beneficial that stone remains available in active, or at least preserved, quarries to allow its use in the reconstruction and restoration of historic buildings and monuments. Compilation and dissemination of information are, together with the preservation of stone quarries, some of the main objectives within this subject, and one way of preserving a quarry when not in use is to give recognition to the stone that at some point in the past

played an important role in building the heritage around the world that we see today. Making the research groups more inclusive will widen the diversity that we all should look for in the heritage stones community. A wide scientific community around the world has expertise in stone characterization and they should join the existing working groups (e.g., HSS) to add information about stones from unrepresented areas, thus helping in the technical and scientific characterization of stones, preserving humanity's stone-built heritage, and improving naming to prevent the use of incorrect stone in restoration projects. Through implementing these recommendations, the original goals of the heritage stones working group would be fully met.

**Funding:** This research received no external funding.

**Data Availability Statement:** The data presented in this review are openly available at http://diarium.usal.es/mdp (accessed on 10 April 2023) or can be accessed via a request to the author.

**Acknowledgments:** The author wants to acknowledge the work of many scientists who characterized and published data on stones, without which this research area would have remained ignored by the global community.

**Conflicts of Interest:** The author declares no conflict of interest.

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
