# Peer review of "The Value of Natural Stones to Gain in the Cultural and Geological Diversity of Our Global Heritage"

_heritage, doi:10.3390/heritage6060241_

Round 1
Reviewer 1 Report
To the author:
Please consider the following:
Lines 48 to 59: explain the goals and achievements of both groups when addressing them for the first time. It becomes clearer to the reader.
Line 56: An updated website is probably not the best way to assess a group's work. Substantiate this sentence with other metrics or remove the sentence.
Lines 62 to 73: If all this information is necessary (which I believe it is not) it is presented poorly, with confusing chronology mixed with too many acronyms. Consider explaining the most relevant relationships and dates within a figure.
Line 77: was not English
Line 79: started a list
Line 94: all information is now available at:..... Remove text addressing the shift to a personal website. Provide reason for this shift, which is not clear in the text. Add the information on the website here (lines 117 to 126).
Line 126: not be accessible to (instead of hidden).
Lines 128 to 133: this is one of the goals of the study and should be presented along lines 161-167.
lines 133 to 143: this should go into the "Discussion" section.
Line 150: is it HRS now?
Lines 153 to 159 : this should go into the "Discussion" section.
Line 171 to 176: add clarity to this section. The methodology should end here, including the presentation of Table 1. See comments on line 288 for a suggested second table. After the table, the author is analysing data so it should fall into the Discussion section.
Table 1:explain to the reader why some entries in the table are numbered.
Line 177: Discussion section starts here with a brief recap on the tables followed by the diversity analysis (3. Discussion, 3.1 Geographic Diversity...)
Lines 243 to 284: this makes more sense in another part of the article, not between two sections on diversity (geographical and lithological).
Line 288 suggests a second Table (12 out of 32) can be presented as well, with the most recent candidates and attributions.
The text presently under 4. Discussion should be connected to parts of the article that are now placed under 3. Diversity because they all discuss , interpret and comment on the data presented in the 2. Methodogy section (where only an introductory text and the table(s) should be). This way, redundancies can be avoided. It is very difficult to foresee how the article will look like after these modifications have been made and if additional changes are needed after restructuring it.
The article is interesting, well-written and relevant but needs severe re-structuring.
The English is good and demands only minor corrections.
Author Response
Dear reviewer,
Thank you very much for the detailed review and the suggestions to improve the paper. I have tried to combine the suggestions and advice from reviewers. believe I have followed all your advice and I explain here how it was done:
- Lines 48 to 59: explain the goals and achievements of both groups when addressing them for the first time. It becomes clearer to the reader.
This has been done by adding the goals and achievements that both groups have explained, either in their web-sites or publications.
- Line 56: An updated website is probably not the best way to assess a group's work. Substantiate this sentence with other metrics or remove the sentence.
This has been reworded
- Lines 62 to 73: If all this information is necessary (which I believe it is not) it is presented poorly, with confusing chronology mixed with too many acronyms. Consider explaining the most relevant relationships and dates within a figure.
I have tried to explain in detail, including the full names of groups and institutions. I have also included a diagram as Figure 1 to explain the different steps and evolution of the working group, since its creation in 2008, its recognition by IUGS in 2012 until today.
- Line 77: was not English
Changed
- Line 79: started a list
Changed
- Line 94: all information is now available at:..... Remove text addressing the shift to a personal website. Provide reason for this shift, which is not clear in the text. Add the information on the website here (lines 117 to 126).
Changed
- Line 126: not be accessible to (instead of hidden).
Changed
- Lines 128 to 133: this is one of the goals of the study and should be presented along lines 161-167.
Changed
- lines 133 to 143: this should go into the "Discussion" section.
Changed
- Line 150: is it HRS now?
It has never been HRS. The subcommission is called at present Heritage Stone Subcommission (HSS). I checked through the paper to make sure there was not misspellings.
- Lines 153 to 159 : this should go into the "Discussion" section.
Changed
- Line 171 to 176: add clarity to this section. The methodology should end here, including the presentation of Table 1. See comments on line 288 for a suggested second table. After the table, the author is analysing data so it should fall into the Discussion section.
The Methodology section has been cleaned as suggested. An extra table would be too repetitive, so instead I have included some details and rearrange the section, moving the analysis of data to Discussion, as suggested.
- Table 1:explain to the reader why some entries in the table are numbered.
The numbers were meant to help on counting the different countries, but I think there was no need for this information, so I deleted the column of numbers.
- Line 177: Discussion section starts here with a brief recap on the tables followed by the diversity analysis (3. Discussion, 3.1 Geographic Diversity...)
Changes have been applied to follow advise and suggestions.
- Lines 243 to 284: this makes more sense in another part of the article, not between two sections on diversity (geographical and lithological).
Those lines have been moved out to a proper place at the end of the diversity subsections.
- Line 288 suggests a second Table (12 out of 32) can be presented as well, with the most recent candidates and attributions.
I have added the web site where the lithologies can be found. There is very little new information for another table and it would look redundant.
- The text presently under 4. Discussion should be connected to parts of the article that are now placed under 3. Diversity because they all discuss , interpret and comment on the data presented in the 2. Methodogy section (where only an introductory text and the table(s) should be). This way, redundancies can be avoided. It is very difficult to foresee how the article will look like after these modifications have been made and if additional changes are needed after restructuring it.
The text has changed quite a lot when re-structuring, as suggested. Hopefully I have followed all the suggestions and now it looks better and more comprehensive.
Reviewer 2 Report
The paper: The value of natural stones to gain in the cultural and geological diversity of our global heritage is aimed to disseminate the great commitment of the HSS group, its history, its birth and the work carried out, giving a data base of the stones used in the Cultural Heritage in the world. The group finds and tries to collect information about beautiful and unknown stones, stimulate study and characterization. Nevertheless, the paper does not seem to underline captivatingly these aspects. I would suggest to put more emphasis on the characteristics that should be studied for each stone, to report the studies carried out until today, to underline the need to discriminate always between a commercial and a scientific nomenclature. It seems that everything is very vague and many things are always repeated. In detail I give some suggestions:
Introduction line 56-58: it is not clear: the group is not active but the literature is increased, from who?
Introduction Line 62-82 : I suggest to insert an explanation of the different acronyms that have been put because it is difficult to understand
Introduction line 79 I suggest to delete “somewhat” before experts
Introduction line 94: IGCP-637 what does it mean? why 637?
Introduction line 98: GHSRs another acronym, I suggest a table with a better explanation on the succession of the various groups.
Introduction line 133-140: this part is interesting and articulated, however, it does not fit well in the introduction but should be placed as an example in a subsequent phase of the paper (for example the discussion).
Methodology line 168 : which is the methodology used? it is not clear, I suggest to write better about the criteria used
Diversity line 199: after 18 I think is “are from Europe” not “where from Europe”
I suggest to put Table 3 before Figure 4
Diversity: Table 2 is not centered
Diversity Figure 3 is not centered and in Figures 3 you must put letter (a) for 2020 and (b) for 2022 and recall them in the caption.
Diversity line 227-234: the explanation of “other” must be put in the text not in the caption of the Figure 4.
Diversity line 243: I suggest to put as subject the stones correlated to the Geo parks and not vice versa
Diversity line 260: I suggest to put the description of the Vietnamese marble in the discussion and not here
Chapter 3.2 Lithology diversity in stones listed and recognized by the IUGS: I suggest to change the title in for example “Problems related to the incorrect naming of stones”
Discussion: in my opinion in this chapter would be appropriate to include all the existing problems with related examples, see marble in Vietnam, lack of scientific characterization, problems of incorrect naming, etc.
Conclusion line 399: I suggest to remove “major” before conclusion; also in this chapter the discussion topics should be resumed
The English linguage is understable and correct, in my opinion it is necessary a moderate editing
Author Response
Dear reviewer,
Thank you very much for the detailed review and the suggestions to improve the paper. I have tried to combine the suggestions and advice from reviewers. believe I have followed all your advice and I explain here how it was done:
- The paper: The value of natural stones to gain in the cultural and geological diversity of our global heritageis aimed to disseminate the great commitment of the HSS group, its history, its birth and the work carried out, giving a data base of the stones used in the Cultural Heritage in the world. The group finds and tries to collect information about beautiful and unknown stones, stimulate study and characterization. Nevertheless, the paper does not seem to underline captivatingly these aspects. I would suggest to put more emphasis on the characteristics that should be studied for each stone, to report the studies carried out until today, to underline the need to discriminate always between a commercial and a scientific nomenclature. It seems that everything is very vague and many things are always repeated. In detail I give some suggestions:
A full paragraph with the main features that should be studied for each stone is included, referring also to a paper that was published with most needed information, and also a figure/diagram to understand the evolution of the working group.
- Introduction line 56-58: it is not clear: the group is not active but the literature is increased, from who?
Those sentences have been reworded. The group seems to be inactive in the last years, but other groups and individuals have published papers on the mentioned neglected areas.
- Introduction Line 62-82 : I suggest to insert an explanation of the different acronyms that have been put because it is difficult to understand
All the acronyms are accompanied by full explanations. A diagram (Figure 1) has been added to help in understanding the structure of the subcommission within the IUGS.
- Introduction line 79 I suggest to delete “somewhat” before experts
Deleted
- Introduction line 94: IGCP-637 what does it mean? why 637?
IGCP is a very prestigious UNESCO program for Earth Science. The number relates to the project and depends on the approval date, increasing the numbers every year. A paragraph explaining UNESCO IGCPs has been added.
- Introduction line 98: GHSRs another acronym, I suggest a table with a better explanation on the succession of the various groups.
All acronyms are accompanied by full explanation.
- Introduction line 133-140: this part is interesting and articulated, however, it does not fit well in the introduction but should be placed as an example in a subsequent phase of the paper (for example the discussion).
That part has been moved
- Methodology line 168 : which is the methodology used? it is not clear, I suggest to write better about the criteria used
The Methodology section has been cleaned, focusing on the methods used to prepare the paper.
- Diversity line 199: after 18 I think is “are from Europe” not “where from Europe”
Typo has been corrected
- I suggest to put Table 3 before Figure 4
Changed
- Diversity: Table 2 is not centered
Changed
- Diversity Figure 3 is not centered and in Figures 3 you must put letter (a) for 2020 and (b) for 2022 and recall them in the caption.
Figure has been centered. As each figure has a year above, I think there is no need to include a and b. However, in the legend, above and below have been included.
- Diversity line 227-234: the explanation of “other” must be put in the text not in the caption of the Figure 4.
Changed
- Diversity line 243: I suggest to put as subject the stones correlated to the Geo parks and not vice versa
Changed, as suggested
- Diversity line 260: I suggest to put the description of the Vietnamese marble in the discussion and not here
I think this description is better explained in this part, but some arrangements have been done to make more sense of this description.
- Chapter 2 Lithology diversity in stones listed and recognized by the IUGS: I suggest to change the title in for example “Problems related to the incorrect naming of stones”
I have not changed the subtitle, but I have included the suggested sentence in this subsection.
- Discussion: in my opinion in this chapter would be appropriate to include all the existing problems with related examples, see marble in Vietnam, lack of scientific characterization, problems of incorrect naming, etc.
The suggested ideas have been included by moving part of the text from Introduction to Discussion.
- Conclusion line 399: I suggest to remove “major” before conclusion; also in this chapter the discussion topics should be resumed
Changed and followed suggestions.